# The Effectiveness of Interventions Delivered Using Digital Food Environments to Encourage Healthy Food Choices: A Systematic Review and Meta-Analysis

**DOI:** 10.3390/nu13072255

**Published:** 2021-06-30

**Authors:** Rebecca Wyse, Jacklyn Kay Jackson, Tessa Delaney, Alice Grady, Fiona Stacey, Luke Wolfenden, Courtney Barnes, Matthew McLaughlin, Sze Lin Yoong

**Affiliations:** 1School of Medicine and Public Health, College of Health, Medicine and Wellbeing, University of Newcastle, Callaghan, NSW 2308, Australia; Jacklyn.Jackson@health.nsw.gov.au (J.K.J.); Tessa.Delaney@health.nsw.gov.au (T.D.); Alice.Grady@health.nsw.gov.au (A.G.); Fiona.Stacey@health.nsw.gov.au (F.S.); Luke.Wolfenden@health.nsw.gov.au (L.W.); Courtney.Barnes@health.nsw.gov.au (C.B.); Matthew.Mclaughlin1@health.nsw.gov.au (M.M.); Serene.Yoong@health.nsw.gov.au (S.L.Y.); 2Hunter Medical Research Institute (HMRI), New Lambton Heights, NSW 2305, Australia; 3Priority Research Centre for Health Behaviour (PRCHB), University of Newcastle, Callaghan, NSW 2308, Australia; 4Hunter New England Population Health, Hunter New England Local Health District, Wallsend, NSW 2287, Australia; 5School of Health Sciences, Swinburne University of Technology, Hawthorn, VIC 3122, Australia

**Keywords:** digital food environments, food choice, online intervention, choice architecture, systematic review

## Abstract

Digital food environments are now commonplace across many food service and retail settings, influencing how the population orders and accesses foods. As such, digital food environments represent a novel platform to deliver strategies to improve public health nutrition. The purpose of this review was to explore the impact of dietary interventions embedded within online food ordering systems, on user selection and purchase of healthier foods and beverages. A systematic search of eight electronic databases and grey literature sources was conducted up to October 2020. Eligible studies were randomized controlled trials and controlled trials, designed to encourage the selection and purchase of healthier products and/or discourage the selection and purchase of less-healthy products using strategies delivered via real-world online food ordering systems. A total of 9441 articles underwent title and abstract screening, 140 full-text articles were assessed for eligibility, and 11 articles were included in the review. Meta-analysis of seven studies indicated that interventions delivered via online food ordering systems are effective in reducing the energy content of online food purchases (standardized mean difference (SMD): −0.34, *p* = 0.01). Meta-analyses including three studies each suggest that these interventions may also be effective in reducing the fat (SMD: −0.83, *p* = 0.04), saturated fat (SMD: −0.7, *p* = 0.008) and sodium content (SMD: −0.43, *p* = 0.01) of online food purchases. Given the ongoing growth in the use of online food ordering systems, future research to determine how we can best utilize these systems to support public health nutrition is warranted.

## 1. Introduction

Diet is a major modifiable risk factor for disease morbidity and mortality [1]. In 2019, approximately eight million deaths worldwide were attributed to dietary risk factors (including insufficient consumption of fruits, vegetables, wholegrains/legumes, and excessive consumption of sodium, fat, sugar and energy), predominantly from cardiovascular disease, cancer and diabetes [2]. Although most dietary guidelines recommend the consumption of at least five servings of fruit and vegetables per day for the prevention of chronic diseases, international data suggest that 72–95% of the population in Europe [3], United Kingdom (UK) [4], United States (US) [5] and Australia [6] fail to meet these recommendations. Population studies from these regions also indicate that dietary intakes are consistently high in energy [7], fat [8], sugar [9] and sodium [10], placing a large proportion of the population at an increased risk of poor health and non-communicable diseases (NCDs) [7].

The food environment is widely acknowledged as a key factor shaping population food decisions and as a driver of NCDs internationally [11]. In recent years, food environments have undergone rapid changes. Increased broadband access and improved safety of electronic payments combined with a rising demand for convenient meal options represent major facilitators driving the evolution of digital food environments and growth as a market sector [12,13]. Digital food environments are broadly defined as electronic interfaces through which people interact with the wider food system [14], and allow consumers to electronically order food and beverages either for pick-up or home delivery [12]. These digital food environments are inclusive of online food delivery (OFD) services (e.g., Uber Eats, Zomato, Grubhub), online canteens and cafeterias (e.g., in schools, workplaces and hospitals), online supermarkets, online meal and food subscriptions (e.g., HelloFresh, Green Chef, Dinnerly), and pre-order mobile applications (e.g., Starbucks pickup) [12,15].

Online and mobile food ordering platforms are now routinely integrated across many different food settings and have had a major impact on the way we can select and access food. In 2020, OFD services alone reached over 935 million users across the US, UK, Europe and China, with approximately 60% of OFD users accessing these services at least once a month [16]. Additionally, with recent COVID-19 lockdowns enforced across major metropolitan areas internationally, there has been a mass of new users to the sector [17], allowing online food ordering use to reach a reported all-time high [18]. As such, online food ordering systems not only represent a key aspect of the contemporary food environment, but they also represent a potential platform to deliver behavioral strategies to support public health nutrition to a large number of people, on a regular basis, at a key behavioral moment (the point-of-purchase), and at a relatively low cost.

Within many food environments, availability, accessibility, affordability, as well as media and advertising have been identified as the primary determinants of food purchasing behavior [19]. Interventions that address these factors within food environments are often based on ‘Choice architecture’ principles. These principles seek to structure environments in ways that automatically promote healthful decisions, rather than relying on the consumer to make effortful and deliberate decisions [20]. Examples of these strategies may include adding interpretive nutrition labels to food items (i.e., making information visible); making the healthy options the most visible and easiest to access (i.e., changing option related effort); and using prompts to ‘nudge’ consumers to make healthy choices (i.e., changing choice defaults). For example, choice architecture strategies have been tested as part of a two-phase choice architecture intervention to improve healthy food and beverage choices within a large ‘bricks and mortar’ hospital cafeteria [21]. As part of the first phase, a color-coded traffic-light labeling system (green = healthy, yellow = less healthy, red = unhealthy) was applied to all foods and beverages, and led to a decrease in total unhealthy sales (−9.2%, *p* < 0.001), and an increase in total healthy sales (+4.5%, *p* < 0.001) from baseline [21]. As part of phase two, the refrigerator and store shelves were rearranged so that healthy items were located at eye-level, and less healthy and unhealthy items were located below eye-level, a strategy that led to a further 4.9% decrease in total unhealthy sales, and large increase (25.8%) in healthy beverage (i.e., bottled water) purchases [21]. Additionally, there is now a growing body of systematic review evidence from physical food environments that choice architecture strategies can improve the food selection and purchases of consumers. For example, product placement in supermarkets (whereby healthy items are placed in prominent positions, and unhealthy items are placed in less prominent positions) has been shown to encourage healthier consumer choices [22]. The addition of shelf labelling interventions (supported by posters and booklets) have also been shown to improve the healthiness of sales in physical supermarkets [23]. As such, these review findings indicate that in some food service and retail settings, subtle and non-invasive strategies can significantly influence consumer purchasing decisions about food and beverages [20,22,23].

There remains a significant opportunity to use established digital food environments as a novel means to deliver such public health nutrition interventions [12,15,24]. However, for these interventions to have a real-world impact, interventions delivered via existing online food ordering systems must be established as effective at promoting good nutrition [25,26]. The effectiveness of choice architecture intervention strategies within real-world digital food environments has not yet been rigorously synthesized. This study aimed to address this knowledge gap by conducting a systematic review to explore the impact of dietary interventions embedded within online food ordering systems on user purchasing of healthier foods and beverages. The secondary objectives of this review were to identify any unintended adverse consequences (e.g., an increase in unhealthy sales), and describe the cost and cost effectiveness of the included interventions.

## 2. Materials and Methods

This systematic review was conducted and reported in accordance to the Preferred Reporting Items for Systematic reviews and Meta-Analyses (PRISMA) guidelines [27]. A review protocol was prospectively published on the Open Science Framework on the 28 September 2020 [28].

### 2.1. Study Selection Criteria

#### 2.1.1. Types of Studies

Study designs eligible for inclusion in the review included parallel group randomized controlled trials (RCTs), cluster-RCTs, stepped-wedge RCTs, factorial RCTs, multiple baseline RCTs, randomized controlled crossover trials, quasi-randomized controlled trials, and non-randomized clinical controlled trials (CCTs). Comparative effectiveness trials were eligible for inclusion provided the study also included an appropriate comparison group.

Observational studies and experimental trials that did not have a comparison group, were excluded from the review. Published and grey literature full-text articles including dissertations and theses were eligible for inclusion; however, conference abstracts without an associated full-text article were excluded from the review.

#### 2.1.2. Type of Participants

This review included studies with participants that were generally healthy, and participants of any age group. Included studies could target any user of an online food ordering system operationalized within a real-world digital food environment. Studies that targeted populations with specific conditions (e.g., hypertension or diabetes), or had a focus on clinical populations (e.g., hospital inpatients) were ineligible, as these health conditions may influence participant food selection or purchases differently, due to a possible need for dietary restriction or adherence.

#### 2.1.3. Types of Interventions

This review sought to include dietary interventions delivered via online food ordering systems operationalized within real-world digital food environments, where an actual online transaction was made in exchange for foods and beverages. As such, interventions were included for review if they met the following criteria:The intervention was delivered primarily via an online food ordering system (i.e., >50% of the intervention strategies were delivered via the online food ordering system). Online food ordering systems of interest included (but were not limited to) online supermarkets and grocery stores, online restaurants, cafes and canteens; and online food and meal delivery services.The intervention aimed to encourage the purchase of healthier foods/beverages and/or reduce the purchase of less-healthy foods/beverages via strategies employed within the online food ordering platform.The intervention involved an actual online transaction, where money or equivalent (i.e., credit/voucher) was directly or indirectly (i.e., in the form of free or subsidized meal programs) exchanged for foods or beverages. This was to ensure that the consumer purchasing behaviors were generalizable to real-world contexts.

Included interventions could also be delivered via any website or mobile-based food ordering platform that allowed consumers to order foods and beverages online. Interventions delivered via OFD services, including Uber Eats (or similar), were also eligible for inclusion.

The online strategies utilized by included interventions were classified according to the following choice architecture techniques as defined by the choice architecture taxonomy developed by Mȕnscher et al. (2016) [29]:Translating information: translating existing, decision-relevant information by changing the format or presentation of the information, but not the context.Making information visible: making external information, that is normally invisible, visible (e.g., daily calories allowances).Providing a social reference point: role modelling, or referring to the behavior of peer groups.Changing choice defaults: setting no-action defaults, or the use of prompted choice (e.g., nudge).Changing option related effort: changing the physical or financial effort to encourage or discourage certain choices.Changing the range or composition of options: changing categories or changing the grouping of options.Changing option consequences: changing the social consequences of certain decisions, or connecting decisions to benefits or costs (e.g., price promotions or discounts).Providing decision assistance: providing reminders, or facilitating commitment (e.g., self or public commitment).

#### 2.1.4. Types of Comparison

Eligible intervention studies needed to include a comparison group that received either; no intervention (i.e., true control), a delayed intervention (i.e., wait-list control), usual care, or an alternative intervention that did not seek to influence food purchasing behavior, and/or was not delivered using an online food ordering system.

#### 2.1.5. Types of Outcomes

This review included interventions that aimed to encourage healthy and/or discourage unhealthy food and beverage purchases (see Appendix A). As such, studies were eligible for inclusion if they reported outcomes related to food and beverage purchases, using sales/purchase data, or direct observation data. Examples of study outcomes of interest included (but were not limited to):The contents of food/beverage purchases according to food groups (e.g., servings of fruits and vegetables), food categories (e.g., the proportion of ‘healthy’ items and ‘less healthy’ items), or the presence of target items (e.g., sugar sweetened beverages).The macronutrient and micronutrient content of food/beverage purchases (e.g., mean energy, saturated fat, total sugar or sodium; or % energy contributed from fat or sugar; or energy density).

The secondary outcomes of this review were any unintended adverse consequences due to the intervention (e.g., changes in service operations), and any cost (e.g., change is business revenue; or increased costs to the consumer) or cost-effectiveness outcomes related to included interventions.

### 2.2. Search Strategy

A systematic electronic database search was conducted from database inception to 1 October 2020. Electronic database searches were conducted in Medline, EMBASE, PsycINFO, ERIC (Proquest), CINAHL complete, Scopus, Business Source Ultimate, and Informit Business Complete. The search terms used were endorsed by an experienced search librarian and were based on the following domains using Medical Subject Headings (MeSH) for ‘online’ and ‘food/nutrition’ and ‘selection/purchases’ and ‘randomized controlled trials/controlled clinical trials’. The full Medline search strategy can be found as Appendix A). Grey literature searches in Google and Google Scholar (first 100 search results) were also conducted, using the following search string: [online OR internet OR website] AND [nutrition OR fruit OR vegetable OR food service OR menu OR eating OR cafeteria OR diet] AND [purchase* OR sale* OR order* OR select* OR choice*] AND [random* OR trial]. Furthermore, the reference lists of included studies were assessed to identify additional eligible studies. We did not impose any language or time restrictions on the search.

### 2.3. Data Collection and Analysis

#### 2.3.1. Selection of Studies

Pairs of experienced review authors (J.K.J., A.G., T.D., M.M., and C.B.) independently screened titles and abstracts using Covidence software. Discrepancies were resolved by consensus between reviewers, or where there was insufficient detail available to exclude on the basis of study title and abstract, these studies progressed to full-text review. Pairs of reviewers (J.K.J. and A.R.) independently assessed full-text articles for their eligibility for inclusion. Reasons for excluding studies at full-text review were documented (Figure 1. PRISMA flow diagram of included studies). If discrepancies between reviewers for study inclusion could not be resolved by consensus, a field expert (R.W.) was consulted as a third reviewer to determine final study inclusion.

#### 2.3.2. Data Extraction and Management

Pairs of un-blinded reviewers (J.K.J. and F.S. or R.W.) independently extracted the data from included studies using an adapted and piloted version of the Cochrane Public Health data extraction template. Any discrepancies between reviewers were resolved by consensus. Using our template, review authors extracted the following data from included studies:Study characteristics: first author, publication year, country, study design, study aim, funding source and sample size.Participant characteristics: age, gender and ethnicity.Intervention characteristics: provider of the online food ordering platform, food ordering environment (e.g., school canteen, restaurant, or supermarket), intervention description, intervention strategies (as per Mȕnscher et al. [29] Choice architecture taxonomy), duration and intensity of the intervention.Outcome characteristics: definitions, methods of outcome assessment, and time points of outcome measurements.Study results relevant to the review primary outcome: e.g., food and beverage purchases/selection.Study results relevant to the review secondary outcomes: e.g., unintended adverse events, economic data/evaluation.Conflict of interest: using the Tool for Addressing Conflicts of Interest in Trials (https://tacit.one/, accessed on 7 August 2020).

#### 2.3.3. Study Risk of Bias Assessment

Pairs of review authors (J.K.J. and F.S. or R.W.) independently assessed the risk of bias of included RCTs, using the Cochrane Collaboration’s Risk of Bias (RoB) tool [30]. The Risk Of Bias In Non-randomized Studies-Of Interventions (ROBINS-I) assessment tool was used for included CCTs [31], as described in the Cochrane Handbook for Systematic Reviews of Interventions [32]. Any discrepancies between reviewer ratings were resolved via consensus.

The specific domains of RoB assessment for RCTs related to: (a) random sequence generation (selection bias); (b) allocation concealment (selection bias); (c) blinding of participants and personnel (performance bias); (d) incomplete outcome data (attrition bias); (e) blinding of outcome assessment (detection bias); (f) selective reporting (reporting bias); and (g) ‘other’ biases. For cluster-RCTs, we assessed additional RoB criteria related to: (a) recruitment to cluster; (b) baseline imbalances; (c) loss of clusters; and (d) incorrect analyses.

The specific domains of ROBINS-I assessment for CCTs related to: (a) bias due to confounding; (b) bias in the selection of participants into the study; (c) bias in the classification of interventions; (d) bias due to deviations from the intended intervention; (e) bias due to missing data; (f) bias in the measurement of outcomes; and (g) bias in the selection of the reporting results.

We judged RCTs overall RoB as ‘low’, ‘high’ or ‘unclear’. High RoB was assigned to RCTs if one or more of the assessed RoB domains were scored as high risk; unclear RoB was assigned to RCTs if one or more of the assessed RoB domains were scored as unclear but not at high RoB for any domains; and low RoB was assigned to RCTs when all RoB domains were assessed as low. Based on ROBINS-I assessment, we judged CCTs overall RoB as ‘low’, ‘moderate’, ‘serious’, or ‘critical’, by applying the criteria outlined by the ROBINS-I detailed guidance, in that low RoB was assigned to CCTs judged as a low RoB for all domains; moderate RoB was assigned to CCTs judged as low or moderate RoB for all domains; serious RoB was assigned if there was serious RoB assigned to at least one domain, but not critical RoB in any domain; and critical RoB was assigned to CCTs if they were judged to be critical RoB in at least one domain [31].

#### 2.3.4. Data Synthesis

Where at least three studies reported sufficiently similar outcomes in adequate detail to enable analysis, we conducted a random-effects meta-analysis on continuous outcomes. We used the inverse-variance method in Review Manager 5.4 to generate standardized mean differences (SMD) and corresponding 95% confidence intervals (95% CI). Separate meta-analyses were conducted to explore the effectiveness of interventions delivered via online food ordering systems on reducing the energy, fat, saturated fat and sodium content of food and beverage purchases. Dichotomous outcomes were not combined in meta-analysis, given there were too few (less than three) similar dichotomous outcomes to combine. These findings were included in narrative synthesis.

A random-effects model for combining data was selected as the most appropriate method for analysis, as it was anticipated that there could be natural heterogeneity among studies attributable to the different doses, durations, populations, settings and intervention strategies. The results from the included cluster RCTs were combined with individually randomized studies provided the reported results had accounted for the cluster design, otherwise we adjusted the effective trial sample size, using the intra-class correlation coefficient reported in the manuscript. The inclusion of data from crossover-RCTs was also deemed appropriate for meta-analysis; however, where crossover trials included multiple relevant intervention arms, we only included data from one intervention arm that best-fit our review aim (i.e., the more comprehensive online intervention). Additionally, where included RCTs studied multiple relevant interventions arms (i.e., multiple intervention arms applied changes to the online food ordering environment) compared with a single comparison arm, we combined the effect estimates across intervention arms to create a single pair-wise comparison as recommended in the Cochrane Handbook [33].

Heterogeneity was explored by examining the forest plots from meta-analyses to visually determine the level of heterogeneity (in terms of the size or direction of treatment effect) between studies. We regarded substantial or considerable heterogeneity as T^2^ > 0 and either I^2^ > 30% or a low *p* value (<0.10) in the chi^2^ test. Sources of possible heterogeneity were explored by sub-group analyses (e.g., all strategies delivered within online food ordering system vs. online and offline strategies), where data from more than three studies had been pooled. Additionally, we carried out sensitivity analyses to examine the effect of removing trials at high risk of bias (based on RoB) or serious/critical risk of bias (based on ROBINS-I assessment) from the analysis. Given our meta-analyses included fewer than 10 studies, we did not explore the risk of publication bias via funnel plot asymmetry due to a lack of power to distinguish chance from real asymmetry [32].

Where outcomes related to the review’s primary outcome were too heterogeneous or reported too infrequently (less than three studies) to be combined using the meta-analysis methods described above, intervention effects were described narratively. Intervention effects for outcomes related to the review’s secondary outcomes were also narratively described.

## 3. Results

### 3.1. Study Selection

The electronic and grey literature search identified a total of 9441 records that underwent title and abstract screening (see Figure 1 for PRISMA flow diagram). Of these, a total of 140 full-text articles were reviewed for eligibility, and 11 studies (all published in peer-reviewed journals) were identified as eligible for inclusion in the review. Seven of these studies were included in a meta-analysis of energy, and three studies each were included in a meta-analyses of fat, saturated-fat and sodium.

### 3.2. Study Characteristics

#### 3.2.1. Design

Six studies were RCTs [34,35,36,37,38,39], two were crossover RCTs [40,41], another two were cluster RCTs [42,43], and one was a non-randomized controlled trial [44]. See Table 1 for characteristics of included studies for further information.

#### 3.2.2. Setting

The online food environments in which studies were conducted included online supermarkets (*n* = 6) [34,35,36,40,41,44], online school canteens and cafeterias (*n* = 3) [37,42,43], and online workplace cafeterias (*n* = 2) [38,39].

Four included studies were conducted in the US [34,37,38,39], four in Australia [36,42,43,44], and the remaining three were conducted in Singapore [35,40,41].

#### 3.2.3. Participants

Studies reported participant sample sizes ranging from 26 [38] to 2371 participants [42]. Of the eight studies including adult participants [34,35,36,38,39,40,41,44], the reported mean age of study participants ranged from 35 years to 47 years, and most (*n* = 6) included a higher proportion of females to male participants. One study recruited food insecure participants, based on their use of a local food pantry [34].

Three included studies were aimed at improving the dietary behaviors of primary school aged children [37,42,43]. In two of these studies, the online school canteen system user may have included the parents of the primary school age children [42,43], however user demographics were not collected as part of these cluster RCTs.

#### 3.2.4. Interventions

Based on the classification of intervention strategies according to Mȕnscher’s [29] choice architecture taxonomy, four studies incorporated only one choice architecture strategy into the online food ordering system [39,40,43,44]. While most other studies incorporated two choice architecture strategies [34,35,36,37,38,41], Delaney et al. incorporated four choice architecture strategies within the online food ordering system [42].

The most commonly used intervention strategy involved applying labels to items within the online menus or product lists to encourage healthier purchasing behaviors. Six interventions applied labels to online grocery store items [35,40,41,44] or workplace menus [38,39], and one intervention applied labels to online school canteen menus [42]. Although most of the labels were applied to make nutrition information visible to study participants, one trial applied a temporary price rise label to high calorie products (real and fake) as a strategy that sought to disincentivize the purchasing of high calorie products by changing the option consequences [35]. Five included interventions sought to influence food purchasing behavior by changing choice defaults (e.g., providing consumers a healthy pre-filled online grocery cart) [34,36,38] or applying prompted choice strategies (e.g., nudging consumers to swap selected items for more healthy options) [37,42]. Two interventions changed the range or composition of options by redesigning the online canteen menus to display healthy menu items (e.g., fruit and vegetable snacks) in more prominent positions [42,43]. Three studies included additional intervention components that were not delivered via the online food ordering platform [38,39,42]. These off-line strategies included providing the food service with feedback on how to improve the availability of healthy foods on their menus [42], providing participants with face-to-face mindful-eating training [38], and emailing participants (external to the online food ordering system) to remind them of the study and eligibility for a discount on food purchases made via the online food ordering system [39].

#### 3.2.5. Comparison Group

The comparison group in nine of the studies was a no intervention control [35,37,38,39,40,41,42,43,44]. Two studies provided an alternate intervention involving nutrition education and the provision of nutrition resources delivered online [34,36], but not via the online food ordering system.

#### 3.2.6. Primary Outcomes

All primary outcomes of interest (i.e., food selections and/or purchases) were collected using objective purchasing data recorded via the online food ordering systems. Seven studies reported on multiple outcomes of interest, related to the healthiness of consumer purchases. Eight of the included studies measured outcomes related to the nutritional content of purchases [34,35,36,38,39,40,41,42], and seven of these measured the energy (calorie or kilojoule) content of purchases (e.g., average calories per purchase, average calories per serve purchased) [34,35,38,39,40,41,42]. Fat and saturated fat was measured in five studies [34,36,38,41,42], while sodium [34,41,42], sugar [41,42], cholesterol [34] and fiber [34] were less frequently measured. The ‘healthiness’ or ‘quality’ of food purchases was evaluated in four studies [35,41,42,44], using various nutrient scores. Additionally, three studies measured outcomes related to specific food groups, including fruits and vegetables [34,37,43], low fat dairy [37], and wholegrains [37].

#### 3.2.7. Secondary Outcomes

Only six studies reported outcomes related to secondary outcomes of interest (i.e., adverse consequences or cost/cost-effectiveness). Two trials measured the change in the businesses’ weekly revenue due to the application of the intervention [42,43] and four studies evaluated the change in the average costs to the consumer per purchasing occasion [35,36,40,41]. None of the studies reported on outcomes related to the cost of delivering the intervention, the cost-effectiveness of the intervention or unintended adverse events.

### 3.3. Risk of Bias

Figure 2 shows the results of risk of bias assessment for included RCTs. It was unclear whether random sequence generation was adequately performed in three trials [37,39,40], while another trial broke randomization by adding additional participants to the delayed intervention group post the initial randomization [38]. Risk of bias for concealment of allocation sequence was unclear in five trials [34,37,38,39,40]. One trial included a face-to-face component that un-blinded participants to their group allocation [38], and was assessed as at high risk of performance bias, while for another trial this was unclear due to a lack of information [39]. In regard to detection bias, all studies were assessed as low risk given the objective nature of the outcome measure used. Three studies provided insufficient information or reasons for dropout at follow-up to determine risk of attrition bias [35,38,39]. Six studies provided sufficient information to be assessed as a low risk of bias for selective outcome reporting [35,36,38,40,41,42]. No ‘other’ sources of bias were determined by RoB assessors.

Both cluster RCTs [42,43] were assessed to have a low risk of bias for recruitment to cluster, loss of clusters and incorrect analysis. One cluster RCT was assessed high risk of bias for baseline imbalance [43], as some baseline imbalances between groups were not accounted for in the analysis.

The single CCT was assessed as per ROBINS-I (results not included in Figure 2) [44]. Bias due to selection of participants, classification of interventions, measurement of outcomes and selection of reported results was determined as low. Bias due to deviations from the intended intervention and due to missing data was unable to be determined due to insufficient information. Bias due to confounding was assessed as moderate, giving this study a moderate overall risk of bias.

### 3.4. Intervention Effects: Meta-Analysis

#### 3.4.1. Energy Content of Purchases

Seven studies reported on the energy content of purchases, and all could be combined in a meta-analysis to explore differences in the energy content of online purchases [34,35,38,39,40,41,42]. The results (presented in Figure 3) suggest that interventions delivered via online food ordering systems have a moderate and statistically significant effect on reducing the energy content of purchases (SMD: −0.34 [95% CI: −0.60, −0.08], *p* = 0.01). Statistical significance was retained when high risk of bias studies (*n* = 1 [38]) were removed from the analysis (SMD: −0.29 [95% CI: −0.55, −0.03], *p* = 0.03).

Exploratory subgroup analysis according to whether all intervention components were delivered via the online food ordering system [34,35,40,41] (*n* = 4) versus interventions that included strategies both online and offline [38,39,42] (*n* = 3), indicated that there was a statistically significant subgroup effect (*p* = 0.03). Interventions delivered via online food ordering systems that included strategies delivered offline led to a greater reduction in the energy content of food purchases (SMD: −0.63 [95% CI: −1.03, −0.24], *p* = 0.002) (Appendix B. Figure A1). However, there was substantial unexplained heterogeneity between the trials within each of these subgroups (online only: I^2^ = 69%; online plus other: I^2^ = 69%).

#### 3.4.2. Total Fat and Saturated Fat Content of Purchases

Three studies reported data that could be pooled to explore differences in the fat content of online purchases due to the intervention delivered via online food ordering systems [34,38,41]. The results are presented in Figure 4. Pooled analysis suggests that the included interventions significantly reduced the fat content of online purchases (SMD: −0.83 [95% CI: −1.60, −0.05], *p* = 0.04). Once any high risk of bias studies were removed from the analysis [38], this effect was no longer statistically significant (SMD: −0.74 [95% CI: −1.76, 0.28], *p* = 0.15); however, given the small number of studies included in this analysis (*n* = 2), these results should be interpreted with caution.

Data related to the saturated fat content of purchases could be combined in meta-analysis for three studies [34,41,42], and are presented in Figure 5. Overall, the results suggest a reduction in the saturated fat content of online purchases as a result of the online intervention (SMD: −0.71 [95% CI: −1.30, −0.12], *p* = 0.02).

#### 3.4.3. Sodium Content of Purchases

Three of the included studies reported sufficient data to be combined in a meta-analysis to explore intervention effects on the sodium content of online purchases [34,41,42] (presented in Figure 6). Overall, the results suggest a reduction in the sodium content of online purchases due to exposure to interventions delivered via online food ordering systems (SMD: −0.43 [95% CI: −0.76, −0.09], *p* = 0.01).

### 3.5. Intervention Effects: Narrative Synthesis

#### 3.5.1. ‘Other’ Nutrient Content of Purchases

The findings from studies influencing ‘other’ key nutrient outcomes are mixed. Huang et al. [36] found that offering consumers the opportunity to swap products high in saturated fat (>1% saturated fat) with those that were low in saturated fat at the online checkout, led to a 66% (95% CI: 0.48 to 0.84; *p* < 0.001) reduction in the amount of saturated fat in the food purchased by the intervention group compared with the control group. Similarly, the prefilled (default) online grocery cart intervention by Coffino et al. was found to increase the mean fiber content of purchases by 15.65 mg (95% CI: 3.87, 27.43), and lower the cholesterol content of purchases by 463.86 mg (95% CI: −728.96, −198.76), compared with control participants receiving nutrition education [34]. Finkelstein et al. (2019) [41], found that the application of nutrition labels to online grocery store products did not produce statistically significant differences in the fiber or sugar content of purchases compared with control. Delaney et al. [42] also found that a multi-strategy intervention delivered via online school canteens had no effect on the sugar content of online purchases compared with control schools.

#### 3.5.2. ‘Healthiness’ or ‘Nutritional Quality’ of Purchases

The results from studies assessing the purchase of food groups are also mixed. Coffino et al. [34] (default grocery cart intervention), reported increases in the serves of healthy foods groups, including a 1.5 serve/day increase in fruits (95% CI: 0.59, 2.51), a 2.21 serve/day increase in vegetables (95% CI: 0.41, 4.01), and 4.05 serve/day increase in wholegrain purchases (95% CI: 1.96, 6.14). Miller et al. [37] (who applied nudges to a school online cafeteria system to encourage the purchase of a balanced meal) also found improvements in the food group content of intervention student lunch purchases, indicating a 5.7%, 8.3% and 4.9% increase in the mean proportion of meals containing vegetables, fruit and low-fat milk, respectively (*p* < 0.0001). However, in a cluster RCT conducted by Wyse et al. [43] that involved repositioning fruit and vegetable snack items within the online menu, the intervention did not significantly influence the proportion of online orders containing a fruit or vegetable snack.

Four studies reported on measures of ‘the healthiness’ or ‘nutritional quality’ of purchases as a result of the interventions delivered via online food ordering systems, of which results were mixed. For example, Sacks et al. [44] found the application of nutrition labels to a sub-sample of supermarket products had no effect on the sales of ‘healthy’ and ‘less healthy’ products. Doble et al. [35] also found that the application of various labels to indicate a 20% price rise on high calorie products in a supermarket had no effect on the diet quality of purchases.

In contrast, Finkelstein et al. (2019) [41] found the application of two different nutrition labels (Multi-Traffic Light labels and Nutri-Score) to all supermarket products led to improvements in the diet quality of consumer shopping carts (measured according to the Alternative Healthy Eating Index), but only Nutri-Score labels were effective at increasing the average Nutri-Score of consumer shopping carts. The application of traffic light labels (combined with other online strategies) to school canteen menus was also found effective at increasing the average proportion of items purchased that were classified as ‘high in nutritional value’ and decreasing the average proportion of items purchased classified a ‘low in nutrition value’ [42].

#### 3.5.3. Cost of Interventions Delivered via Online Food Ordering Systems

Four studies examined the costs of the intervention to the online food ordering system users (i.e., the consumer), and all found that there were no significant differences between the intervention and control [35,36,40,41]. Two studies measured the change in business weekly revenue as a result of applying the online intervention to their online ordering system, and found there was no significant difference between intervention and control business revenue [42,43] (see Table 2).

#### 3.5.4. Unintended Adverse Events

Included interventions did not report on any unintended adverse events.

## 4. Discussion

This systematic review was conducted to explore the effectiveness of dietary behavioral interventions delivered using real-world online food ordering systems that seek to encourage the selection or purchase of healthy foods and beverages. The search identified 11 controlled trials (six RCTs [34,35,36,37,38,39], two crossover RCTs [40,41], two cluster RCTs [42,43] and one CCT [44]) that were eligible for inclusion. The included studies had between 26 and 2371 participants, and were conducted in online supermarkets [34,35,36,40,41,44], online school canteens and cafeterias [37,42,43], and online workplace cafeterias [38,39].

The findings from the meta-analysis of seven studies that assessed the energy content of the foods selected through the online ordering systems indicate that interventions delivered via online food ordering systems were effective in reducing the energy content of consumer purchases, with a moderate effective size (SMD: −0.34). The effect size was maintained when studies with a high risk of bias were excluded from the analysis (SMD: −0.29). Similarly, each of the separate meta-analyses for fat, saturated fat and sodium were statistically significant, favoring the intervention group with a relatively large effect size (SMD: −0.83, −0.71, and −0.43, respectively), yet, when high risk of bias studies were removed from the analysis, the effect for total fat was no longer statistically significant (SMD: −0.74, *p* = 0.15). However, the meta-analyses for fat, saturated fat and sodium each had only three studies included, with a relatively large amount of unexplainable heterogeneity, so it is important that these findings are interpreted with caution.

Given that this review assessed differences in the purchasing patterns of consumers following the introduction of an intervention delivered via an online food ordering system, over half (*n* = 6) of the studies included cost data. Four studies reported the average cost of the order to consumers (i.e., price data) [35,36,40,41] and two assessed revenue [42,43]. However, there was no assessment of the cost-effectiveness of interventions at the individual or provider level. Particularly in light of the promising results of the meta-analyses suggesting that these interventions may be effective at reducing intake of energy, fat and sodium, future research in this field should seek to assess the overall cost of implementing these types of interventions and conduct additional economic analysis. As the interventions included in the current review were already embedded within existing online food ordering systems, they represent a relatively low-cost strategy for improving public health nutrition, and potentially for reducing the future burden of diet-related chronic diseases [45,46,47]. Furthermore, given that these systems automatically record the items purchased and the cost of that item, it is anticipated that collection and analysis of cost-effectiveness data should be simple compared to other behavioural trials.

All strategies were able to be classified by Mȕnscher’s [29] choice architecture taxonomy. The 11 included studies tested a variety of intervention strategies, both in isolation and in combination with other strategies. The most commonly tested strategy was ‘making information visible’ (*n* = 9) [35,36,37,38,39,40,41,42,44], and was operationalised in a variety of ways including: identifying healthy or less healthy options (e.g., labelling as high fat, low calorie); or labelling all options with either traffic lights or nutrient content or both. Furthermore, seven of the studies included interventions which included multiple strategies [34,35,36,37,38,41,42]. As such, the diversity of strategies tested and the combination in which they were combined limits the ability to synthesise the evidence for any single strategy, and future research should seek to address this.

Of the 11 included studies, just three (27%) were given an overall judgement of a low risk of bias [36,41,42], three (27%) were judged at high-risk or moderate-risk of bias [38,43,44], and five (45%) were judged as having an unclear risk of bias due to missing or unclear outcome reporting [34,35,37,39,40]. However, given the objective nature of the outcome measure (i.e., the purchasing data) which was automatically collected by the online ordering system, all of the included studies were assessed as low risk of detection bias.

The findings of this systematic review are in broad agreement with previous reviews which have found positive associations between choice architecture interventions delivered in physical food environments and healthy food choices. For example, the review by Skov et al. [48] explored choice architecture interventions delivered in self-service food settings. Narrative analysis suggested that the strategy of health labelling at point-of-purchase (akin to the “Making Information Visible” strategy in the current review) was associated with healthier food choices in all five of the included studies [48]. Similarly, a recent systematic review of choice architecture interventions within the school food setting found that nudges were positively associated with food selection [49]. Furthermore, a review of 26 studies (across 13 articles) exploring the effect of the choice architecture strategies of ‘salience’ and ‘priming’ nudges among adults, found that interventions that combined ‘salience’ and ‘priming’ were consistently associated with healthier food and beverage choices [50]. In contrast to the current review, all three of these previous systematic reviews synthesised the results narratively due to the heterogeneity of the included studies, and only one review was restricted to real-world studies [49]. The sample size of the included studies tended to be smaller in previous reviews, as the focus on online ordering systems in the current review facilitates access to much larger samples. A further point of differentiation is the use of online food ordering systems in the current review allows for objective recording and storage of sales data, which overcomes previous barriers to the collection of large amounts of high-quality data.

All of the included interventions, but particularly those adopting the strategies of ‘making information visible’, are based on classifying the existing food items according to a specified nutrition classification system. As such, it is important to recognise that the impact of such interventions is influenced by the effectiveness of the classification system employed [51]. Blunt systems, with little ability to discriminate between menu items, will limit the effectiveness of any intervention which is based upon that system [51]. An evaluation or assessment of the labelling systems employed within the included trials was beyond the scope of this review. However, the difficulties of using and applying nutritional guidelines consistently, especially for food items that are not pre-packaged, or without a nutritional information panel are acknowledged as a barrier to implementing these interventions on a wide scale [24]. Additionally, the included studies generally did not include detailed information about the menu items that were available for purchase online (e.g., pre-prepared foods vs. whole foods vs. highly processed food). As such, it was not possible to consider the role of the relative availability of healthy vs. unhealthy items in the purchasing decision. However, this represents an area to focus future research.

This systematic review provides preliminary evidence that dietary interventions delivered via online food ordering systems can influence the purchasing behaviour of consumers. However, their success relies on the willingness and support of the providers of online ordering systems [24], and willingness to adopt these interventions may vary by sector. For example, within the school food service sector (i.e., canteens/cafeterias), there are food standards and guidelines that often govern the provision of food to students. Providers in this setting may be more amenable to adopting these interventions. It should also be noted, however, that although there is not the same level of regulation or guidelines for foods made available in settings for adults, there is increasing demand for healthy eating support [52], and there may be a market share available to those providers that can respond to such demand.

The findings of this review should be considered with respect to its strengths and limitations. Online food ordering is still an emerging field and is yet to have standardised terminology. To counter this, we used an experienced search librarian to map out terms and run the search. We also employed broad terms, and therefore screened a large number of studies. We followed best-practice principles, as outlined in the Cochrane Systematic Review Handbook, including double screening, extracting data, and assessing risk of bias, and we used recommended tools (e.g., Risk of Bias tool) and analysis approaches [32]. There was considerable heterogeneity within the identified studies, and although outcomes enabled meta-analysis, the analyses for fat, saturated-fat and sodium only contained three studies each and as such, results should be interpreted with caution. Additionally, while we identified some incongruence in the statistical significance of an effect for total fat once high risk of bias studies were removed from the analysis, the effect size remained relatively similar. However, the conduct of additional intervention studies in the field will increase the precision of the effect estimates, and may allow exploration of the differences in intervention effects based on user characteristics (e.g., age and sex), the online food service setting (e.g., online restaurants vs. canteens/cafeterias/kiosks vs. supermarkets), and intervention strategies embedded within the online systems (e.g., making information visible vs. changing choice defaults).

There was also a relatively large amount of information that could not be ascertained or was unclear in order to assess risk of bias. However, the included studies were all deemed to be at low risk of detection bias due to the use of routinely collected purchasing data to assess outcomes. The eleven studies included in this review were conducted across only three countries, the US, Australia and Singapore, and only a few online food service settings (e.g., schools, workplaces and supermarkets) which may have implications for the generalisability of our review findings in other countries and across other online food service settings. While the use of real-world systems also addresses issues of selection bias, as the participants were often existing users and in a number of the studies they were not aware they were participating in a research trial. This suggests that the results are more likely to generalise to real-world implementation. This review synthesised evidence for the effectiveness of interventions delivered using real-world online food ordering systems. These systems are already in place and are already used by large numbers of individuals. Although it is unclear whether these results would generalise beyond the users of online ordering systems, it is expected that the pathway to translate this research into practice is minimal compared to virtual or simulated interventions for which substantial work needs to be done to implement into routine practice.

## 5. Conclusions

This is the first review to synthesise evidence of the effectiveness of dietary interventions delivered via real-world online food ordering systems. Despite a relatively small number of studies across a range of settings, meta-analysis suggested that these interventions were effective in reducing the energy, fat, saturated fat and sodium content of online food orders. Given the proliferation of online food ordering systems in recent years and the huge worldwide audience (>1.2 billion), additional research is warranted to determine how best to use these systems to support public health nutrition.

## Figures and Tables

**Figure 1 nutrients-13-02255-f001:**
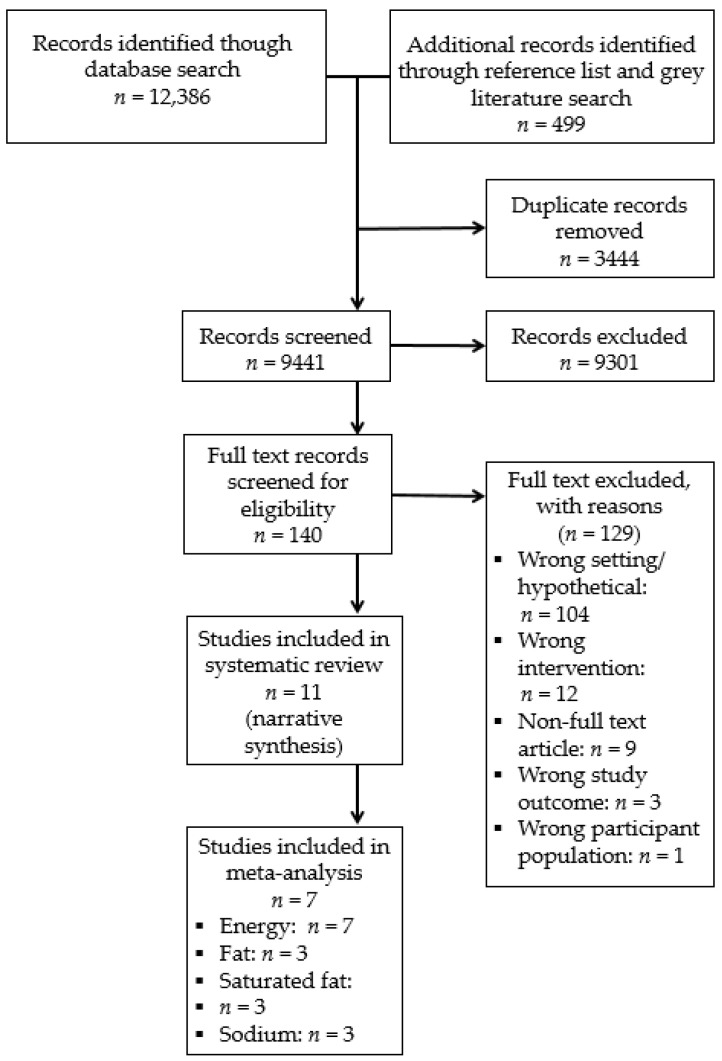
PRISMA flow diagram of included studies.

**Figure 2 nutrients-13-02255-f002:**
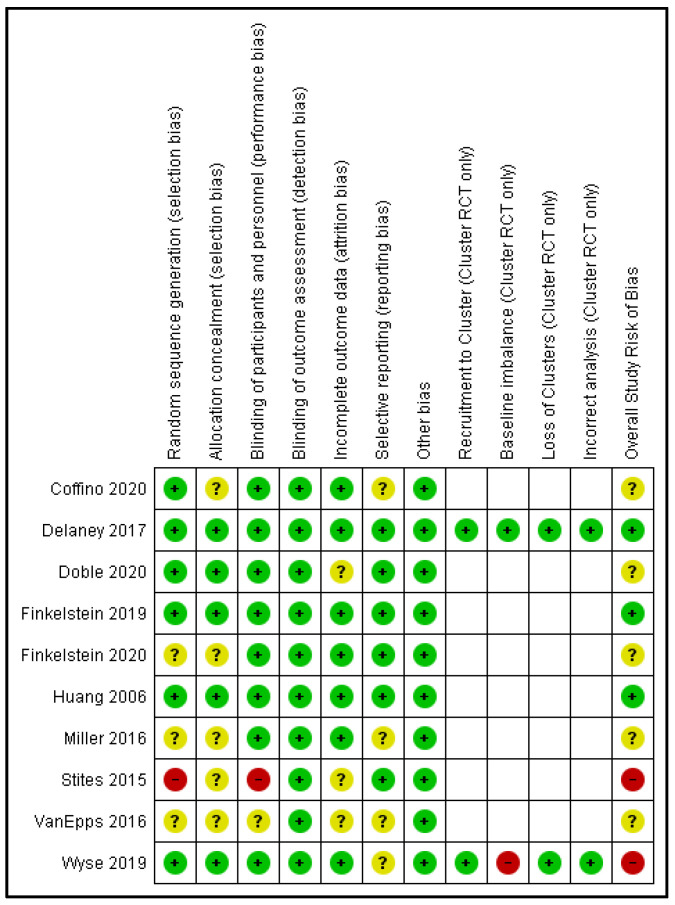
Risk of bias graph: review authors’ judgements about each risk of bias item for each included RCT and cluster-RCT. Green symbol represents low RoB; Yellow symbol represents unclear RoB; Red symbol represents high RoB.

**Figure 3 nutrients-13-02255-f003:**
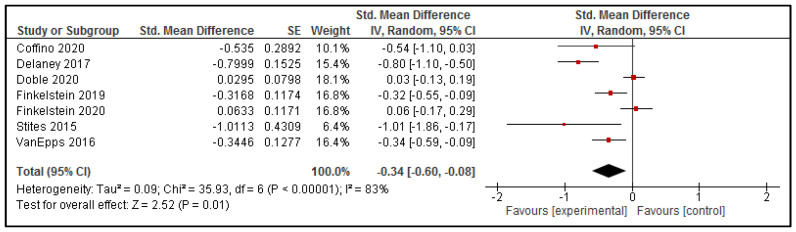
Effect of intervention delivered via online food ordering systems on the Energy content of purchases.

**Figure 4 nutrients-13-02255-f004:**
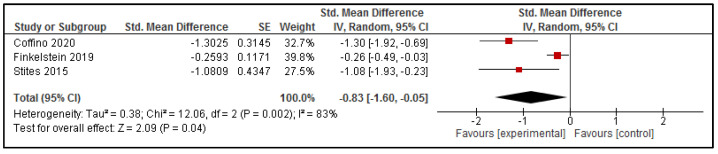
Effect of intervention delivered via online food ordering systems on the Fat content of purchases.

**Figure 5 nutrients-13-02255-f005:**
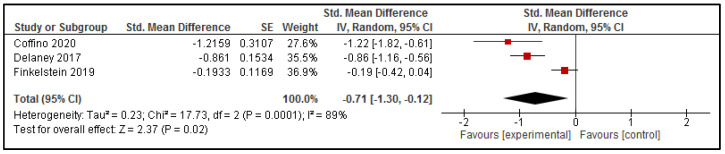
Effect of intervention delivered via online food ordering systems on the Saturated Fat content of purchases.

**Figure 6 nutrients-13-02255-f006:**
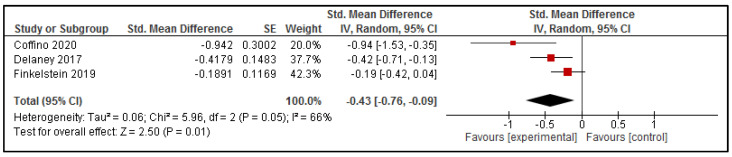
Effect of intervention delivered via online food ordering systems on the Sodium content of purchases.

**Table 1 nutrients-13-02255-t001:** Description of study characteristics.

Author, Year; Study Design, Country	Online Food Ordering Environment; Participant Characteristics	Sample Size *	Intervention Description [Choice Architecture Strategies] ^†^	Duration of Intervention	Control Description	Dietary Outcomes Assessed: Primary Outcomes	Adverse Events/Costs: Secondary Outcomes
Coffino [34], 2020; RCT, US	Online supermarket; Adults (mean age 46.6 years; 76% male) with food insecurity from a single-person household	*n* = 50	Participants were presented with a prefilled online grocery shopping cart containing groceries selected to meet nutrition requirements based on participant sex and age. [Changing choice defaults]. Participants were free to delete, add, exchange or keep items in their cart prior to finalizing their purchase. [Changing option-related effort].	Participants exposed to intervention in a single online grocery shop	**Alternative intervention:** Participants read nutrition education handout prior to completing their purchase.	Average daily food and nutrient content of purchases: Wholegrain (serves/d); Fruit (serves/d); Vegetable (serves/d); Calories (kCal/d); Fat (g/d); Saturated Fat (g/d); Sodium (mg/d); Cholesterol (mg/d); Fibre (mg/d)	Nil
Doble [35], 2020; RCT, Singapore	Online supermarket; Adults (mean age 35.6 years, 49% male), and were Singapore residents.	*n* = 941	20% of products with the highest calories per serving (excluding fruit and vegetables) had a price rise of 20%. There were 3 intervention arms: (1) *Implicit Tax*: High calorie products labelled with ‘raised price’ only (no explaination) (2) *Fake tax*: Shows price pre and post the price rise and falsely indicates that the product will incur a 20% price rise due to high calorie content (3) *Explicit tax:* Shows the same label as fake tax group, but the 20% price rise is actually applied. [Changing option consequences; Making information visible]	Participants exposed to intervention in a single online grocery shop	**True control:** No labels or price manipulation strategy applied	The proportion of taxed (i.e., high calorie) products purchased; The kCal per serve purchased; The Alternative Healthy Eating Index Score of purchases. The kCal per $ spent.	The average total cost per shop.
Finkelstein [40], 2020; Crossover RCT, Singapore	Online supermarket; Adults (mean age 35 years, 21% male), who were the primary shopper for their household and were a registered shopper with RedMart (the online food ordering platform provider)	*n* = 146	For each shop participants could spend between SG$50–250, and asked to complete a typical weekly grocery shop for 3 weeks (3 shops in total). 2 online labelling intervention arms, both applied a ‘Lower calorie’ label: [Making information visible]. (1) *Within category labels applied*: 20% of products within each product category that were lowest in calories per serve were labelled * (2) *Across category labels applied*: 20% of all products that were lowest in calories per serve were labelled.	Participants shopped once a week for 3 weeks (exposed to a different intervention group each week in random order).	**True control:** no online intervention applied to online supermarket.	The proportion of low calorie products purchased per shop. The total calories (kCal) per serve purchased; total calories (kCal) purchased per shop; and total calories purchased per $ spent	Total cost of the shop
Finkelstein [41], 2019; Crossover RCT, Singapore	Online supermarket; Adults (mean age 34.7 years, 31% male), who were Singapore residents and were a registered shopper with RedMart (the online food ordering platform provider)	*n* = 147	For each shop participants could spend between SG$50–100, and asked to complete a typical weekly grocery shop for 3 weeks (3 shops in total). 2 online labelling intervention arms: [Making information visible] (1) *Multiple Traffic Light (MTL)* labels were applied to all products * (2) *Nutri-score (NS)* labels were applied to all products. Prior to each shopping trip, a 60 s video briefly explained how to use the labels that had been applied (MTL or NS). [Translating information]	Participants shopped once a week for 3 weeks (exposed to a different intervention group each week in random order).	**True control:** no online intervention applied to online supermarket.	Diet quality per shopping trip using the AHEI-2010. Average Nutri-Score of the shopping basket, weighted by serve size. The total calories, saturated fat, total fat, sodium and sugar per serve purchased. Calories per $ spent.	Total cost of the shop
Sacks [44], 2011; CCT, Australia	Online supermarket; Customers of online supermarket (participant demographics not further specified)	NA	A set of four traffic light labels to show relative levels of fat, saturated fat, sugar and sodium, were applied to products of the retailer’s own brand (including, milk, bread, breakfast cereals, biscuits and frozen meals) [Making information visible], as there were commercial constrains around labelling branded products. On the home page of the intervention store, and on each of the selected category and product pages, a link was provided to a page providing information about the trial, an explanation of the traffic light indicators, how to interpret them, and general nutrition advice (e.g., Australian dietary guidelines).	Intervention was active for 10 weeks.	**True control:** no nutrition information was provided on the comparison store site during the trial period.	Change in sales by healthiness of products.	Nil
Huang [36], 2006; RCT, Australia	Online supermarket; Adult (mean age 40 years, 12% male) customers of an online supermarket service.	*n* = 456	383 commonly purchased pre-packaged food items that contained >1% saturated fat were selected, and a suitable lower-fat alternative was identified for each (524 foods were identified). Participants assigned to the intervention received advice tailored to the food items they had selected for purchased. This was done automatically by a computer program, and for each items that had >1% saturated fat, participants were presented with the opportunity to retain or swap the item for an alternative, lower saturated fat item (using a side by side comparison of the products). [Making information visible, Changing choice defaults/Prompted choice]	Participants were offered the same form of advice each time that they used the online supermarket during the 5 month recruitment and follow-up period.	**Alternative intervention:** Participants directed to the National Heart Foundation webpage, then prompted to make changes to their purchases.	Mean % saturated fat in the purchased items among the 524 foods studied.	The mean cost per 100 g for the swapped items.
Wyse [43], 2019; C-RCT, Australia	Online School Canteen; Primary schools students (kindergarten to grade 6) attending government schools with an online canteen ordering system. Online canteen user might include parents of schools students	6 schools, and 1903 students	Online canteen menus were redesigned so that fruit and vegetable snack items were positioned first and last on the menu [Changing the range or composition of options]. Target items included fruit or vegetable items (fresh, frozen, tinned or dried) that the children could consume as a snack. Target items were grouped together in a single category titled “fruit and veggie snacks”, which were displayed in 2 places, first and last categories on the online menu. Within this category, items were listed in the following order: whole fresh fruit, cut-up fresh fruit, frozen fruit, tinned fruit, dried fruit, fruit with accompaniments, fresh salad vegetable, cooked vegetables, vegetables with accompaniments.	4 week intervention	**True control:** no changes were made to the online menus.	The proportion of all online orders that contained at least one fruit or vegetable snack food. The proportion of all individual items within all online lunch orders that are a fruit or vegetable snack food.	Average lunch time weekly revenue
Delaney [42], 2017; C-RCT, Australia	Online School Canteen; Primary schools students (kindergarten to grade 6) attending included government schools. Online canteen user might include parents of schools students.	10 schools, and 2371 students.	Intervention schools were provided a canteen menu feedback report to improve the availability of healthy foods (strategy not delivered online). Menu labels (traffic light labels) were applied to online menu items. [Making information visible]. Healthy menu items were listed in the main website display, while users had to click and explore the less healthy menu items. [Changing option related effort]. When users chose unhealthy items, they were prompted to add a healthy item. [Prompted choice]. Healthy items were displayed in bold font, image and positive food prompt “this is a good choice”, [Changing option consequence].	2 month intervention	**True control:** Schools with online canteen did not receive any of the interventions strategies.	The mean energy (kj); saturated fat (g), sugar (g), and sodium (mg) content of student online lunch orders. The mean % energy of student online lunch orders derived from saturated fat and sugar. The mean % of student online lunch orders that were classified as “high nutritional value” and “low nutritional value”.	Canteen weekly revenue
Miller [37], 2016; RCT, US	Online school food service; Elementary or middle school students (5th–6th grade), receiving the National School Lunch Program.	*n* = 71	While pre-ordering lunch online, students received nudges if their meal did not contain all five meal components (i.e., meat/alternative, grain, fruit, vegetables, and dairy). The nudge, “Your meal does not look like a balanced meal” would appear, and a plate was shown as a visual representation and highlight areas of the meal that the student had not selected, and were provided the option to change their orders [Making information visible]. Nudges were primarily provided for fruit, vegetable and/or dairy. Students who selected all 5 components received positive feedback consisting of a smiley face and message “You have ordered a balanced meal”. If a meal remained unbalanced after receiving the nudge, students received a message stating “Please select a fruit, vegetable or dairy. Otherwise you will be charged for each item separately”. Students that did not select these 3 components were charged for each item separately. [Changing option consequence]	Intervention was delivered to students for 2 weeks.	**True control:** Students pre-ordered lunch online, and did not receive nudges. If they had not selected fruit, vegetable or dairy they received: “Please select a fruit, vegetable or dairy. Otherwise you will be charged for each item separately”.	The % of vegetables, fruit and low fat dairy in meals ordered.	Nil.
VanEpps [39], 2016; RCT, US	Online workplace cafeteria; Adults (mean age 40 years, 39% male) employed at a large health care company, and placed at least one online order during the intervention period.	*n* = 249	Using an online based system, participants were required to select exactly one meal for lunch, and had the option to add as many drinks, snacks and desserts that they wanted. There were 13 meal, 23 snack/dessert and 30 drink options. Participants were assigned to 1 of 3 menu labelling options: [Making information visible]. (1) *Traffic light labels* (green, yellow, red: based on calorie content); (2) *Calorie information* appeared next to each menu item. (3) *Combined Traffic light and calorie labelling* On Monday and Wednesday morning, participants received an email reminding them of the study, and the discount (providing in a link to the website).	4 week intervention	**True control:** no online menu labels	Total lunch calories purchased.	Nil.
Stites [38], 2015; RCT, US	Online workplace (hospital) food service; Adult (mean age 44.9 years, 11.5% male) employees who worked full-time at the study hospital and were overweight (BMI > 25 kg/m^2^).	*n* = 26	An online pre-ordering system was developed, to allow participants to order their lunches and view the nutrient content of their choices (calorie and fat content, plus ingredients) [Making information visible]. The system selected the version of the food selected with the least calories and fat by default [Changing choice defaults]. This intervention included Mindful eating training (90 min session) that was delivered offline. 20 × US$1.25 lunch order vouchers were provided to all participants (intervention and control), to encourage the use of the online ordering system.	4 week intervention	**Delayed intervention group**	Average kilocalories and grams of fat in purchased meals.	Nil.

* Represents the intervention arm of the crossover RCT that was included in the meta-analysis. ^†^ Intervention strategies classified according to the Munscher taxonomy [29].

**Table 2 nutrients-13-02255-t002:** Summary of intervention costs.

**Cost of the Intervention to the Consumer**	
Huang, 2006	Mean cost per 100 g of foods purchased: Intervention: AUD $0.63 [0.58–0.68]/100 g Control: AUD $0.62 [0.58–0.067]/100 g
Finkelstein, 2019	Difference in mean total expenditure per shop vs. control: Nutri-Score labels: S$0.90 [SE: 0.98] Multi-Traffic-Light labels: S$1.13 [SE: 1.06]
Finkelstein, 2020	Difference in mean total expenditure per shop vs. control: Within-category labels: S$0.11 [−0.40, 0.63] Across-category labels: S$0.18 [−0.33, 0.70]
Doble, 2020	Difference in mean total expenditure per shop vs. control: Implicit tax: S$1.86 [−1.38, 5.39] Fake tax: −S$0.32 [−3.40, 2.82] Explicit tax: −S$0.79 [−3.83, 2.34]
**Cost of the Intervention to the Foodservice**	
Wyse, 2019	Weekly revenue per school (Relative Mean Difference): AUD $180 [−16, 376], *p* = 0.07
Delaney, 2017	Weekly revenue per school (Relative Mean Difference): AUD −$62.33 [−212.36, 87.68], *p* = 0.41

AUD: Australian Dollar; S$: Singapore Dollar.

## Data Availability

Data is contained within the article or are available from the included studies that have been referenced throughout.

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
