# Peer review of "The Effectiveness of Interventions Delivered Using Digital Food Environments to Encourage Healthy Food Choices: A Systematic Review and Meta-Analysis"

_nutrients, 2021, doi:10.3390/nu13072255_

Round 1

Reviewer 1 Report

This study evaluated the effectiveness of using the online food ordering system for healthy food choices by systematic review and meta-analysis. The authors recruited 11 studies and found that the food ordering system intervention effectively reduces the fat, saturated fat and sodium content. This study has a comprehensive systematic review through search and selection strategies. However, a relatively small number of studies are selected. Also, here are some concerns from the reviewer: 1. Line 485-488: “Once any high risk of bias studies were removed from the analysis, this effect was no longer statistically significant (SMD: -0.74, 95% CI: -1.76-0.28)…..” the interpretation is inappropriate and should discuss more. 2. The purpose of this study was to explore the impact of online food ordering systems on user selection and purchase of “healthier foods and beverages”. However, the main findings of the meta-analysis were energy, fat, saturated fat and sodium content of online food purchases. The authors need to specify the healthier foods, beverages, or unfavorable food selection for defining or searching in this study. Furthermore, is it possible to analyze the impact of the refined sugar content of online food purchases?

Reviewer 2 Report

The current systematic review and meta-analysis investigated whether interventions delivered via an online platform were successful at impacting healthier food choices. Eleven articles were included, and overall results suggested that the variety of online ordering systems included were effective in positively impacting food choices, particularly regarding reduced energy content, fat, saturated fat, and sodium.

The procedures and analyses of the systematic review and meta-analysis were very clear, and the research was well-conducted. The authors are to be commended for their attention to detail. There are certain areas that warrant additional attention, particularly regarding the impact and generalizability of the results. These are detailed in the comments that follow.

  1. Please justify the inclusion of grey literature that may not have undergone sufficient peer-review. In addition, please identify the proportion (if any) of the eleven final articles included that would be considered “grey literature.”
  2. The focus of this manuscript on online food ordering systems is novel, but also restricts the populations that may have been included. For example, use of online ordering systems assumes that the person has access to the necessary technology, and is also capable and comfortable using the technology. This could inadvertently bias the sample against those with lower income, and/or those that do not prefer to use online order systems or do now know how – this could extend to older individuals.
  3. There were three studies that focused on children. Separating these studies from those that only included adults is necessary to determine if there were any differences. There are developmental differences regarding capabilities in decision making. In particular, the children included in the studies were rather young – for these individuals, it is more likely that their parents at least help make decisions about food choices. For the adult samples, the adults are responsible for their own behavior. Given this, it is important to temporarily remove the studies focused on children from the analyses to determine which results are maintained. Then, based upon whether the results change, findings can be discussed while considering developmental differences in the samples.
    Another point here in regards to age, is that the adult samples were mostly focused on adults in the workplace – this is not necessarily representative of all adults across the adult lifespan and thus, the generalizability and implications of this should be considered.
  4. Along with the above point, there was one study included that had participants experiencing food insecurity – further, this sample was dominantly (76%) male, while the remainder of the studies included were primarily female. The recommendation is to temporarily remove this study from the analyses to investigate if the results are impacted. Those experiencing food insecurity may respond differently from those that are not. Further, the gender proportion of the sample is very different.
  5. The result regarding sodium is interesting – of note is that most prepared foods are higher in sodium. If the online order systems included were for prepared foods, this result may have a different meaning than if the studies that led to this result were from online supermarkets. Please consider this in the interpretation of the results and the Discussion.
  6. In the Discussion (p.21) lines 593-603 consider cost. As an additional extension, the economic impact of better food choices would eventually lead to less money spent on health expenditures and chronic disease management – this is positive for individuals, as well as society as a whole from an economic standpoint. This could be integrated in the Discussion as a further implication.
  7. This final comment may be repetitive with some of the above comments. Overall, there is some concern regarding the heterogeneity of the studies included: age, food insecurity, supermarkets versus meals to eat (e.g., workplace cafeterias and school canteen and cafeterias), and location (e.g., country). These sources of heterogeneity should be considered in the Discussion, and to the extent possible, in re-running analyses to see if results change.

Round 2

Reviewer 2 Report

In review of this revised version of the manscript, the authors have sufficiently responded to my prior comments. The authors are to be commended for their careful attention to detail and satisfactory response to the previously mentioned concerns. This revised version of the manuscript appropriately addresses the strengths and limitations of the studies included in the meta-analysis and systematic review so that it provides a clear representation of what conclusions can be made, and provides direction for future research. I have no additional or remaining concerns regarding this manuscript.